# Impact of vitamin D on glycemic control and microvascular complications in type 2 diabetes: A cross-sectional study

**Salma Ahi** [1]*, **Amirreza Reiskarimian**[2], **Mohammad Aref Bagherzadeh**[1,3,4], **Zhila Rahmanian**[1], **Parisa Pilban**[2], **Saeed Sobhanian**[1]

**1** Research Center for Noncommunicable Diseases, Jahrom University of Medical Sciences, Jahrom, Iran, **2** Student Research Committee, Jahrom University of Medical Sciences, Jahrom, Iran, **3** Department of Immunology, School of Medicine, Jahrom University of Medical Sciences, Jahrom, Iran, **4** Department of Advanced Medical Sciences & Technologies, School of Medicine, Jahrom University of Medical Sciences, Jahrom, Iran

* salmaahi.61@gmail.com

## Abstract

Vitamin D has been increasingly recognized for its potential role in modulating various health conditions, including diabetes and its complications. Despite growing evidence suggesting that adequate vitamin D levels may reduce the risk of developing type 2 diabetes and its associated microvascular complications, the precise nature of this relationship remains unclear. This study aims to elucidate the connection among vitamin D status, glycemic control, and microvascular complications in patients with type 2 diabetes, thereby highlighting the importance of vitamin D in diabetes management.This analytical cross-sectional study included 199 type 2 diabetic mellitus (T2DM) patients from the Jahrom city endocrinology clinic. Serum 25(OH)D levels were measured, and their microvascular complications (microalbuminuria, retinopathy, neuropathy, macroalbuminuria) and glycemic control (HbA1C) were measured and confirmed according to ADA guidelines and endocrinologist supervision. All analysis were done with SPSS software. The study enrolled 199 type 2 diabetic patients with a mean age of 56.79 ± 10.8 years, of which 63.3% were female and 57.3% had hypertension. The mean BMI was 28.91 kg/m², and 29.1% of participants had vitamin D deficiency. The prevalence of microvascular complications was 25.6% for retinopathy, 14.1% for neuropathy, and 40% for nephropathy. Vitamin D deficiency was notably higher among patients with retinopathy (37.25%), neuropathy (50%), and macroalbuminuria (56.25%). Patients with neuropathy and retinopathy had significantly lesser serum 25(OH)D concentrations compared to patients without these complications. There was a slight inverse correlation between vitamin D levels and both the urine albumin creatinine ratio (r = -0.175, p = 0.018) and HbA1C (r = -0.19, p = 0.007). Although the link between vitamin D levels and retinopathy was not statistically significant (η = 0.903, p = 0.68), the alteration in vitamin D levels

**Data availability statement:** Due to ethical restrictions imposed by the Ethics Committee of Jahrom University of Medical Sciences to protect participant confidentiality, the data underlying this study cannot be made publicly available. Qualified researchers may request access to the de-identified minimal dataset by contacting the university's independent Ethics Committee at info@jums.ac.ir (with email title: Access to research data) or +98 (715) 4474992. Requests will be reviewed for compliance with ethical standards and institutional regulations. The data will be stored securely in the university's institutional repository, which guarantees long-term preservation and accessibility for approved researchers, irrespective of author availability.

**Funding:** The author(s) received no specific funding for this work.

**Competing interests:** The authors have declared that no competing interests exist.

was suggestively linked with neuropathy ($\eta = 0.975$, $p < 0.001$).Vitamin D deficiency is prevalent among type 2 diabetic patients and is related to a higher occurrence of microvascular complications and poorer glycemic control. These findings underscore the potential importance of managing vitamin D levels in reducing complications and improving diabetes outcomes. Future studies should investigate whether oral vitamin D supplements consumption can improve glycemic control and reduce microvascular complications in these patients.

## Introduction

Healthcare organizations face significant challenges due to the growing burden of chronic diseases, which pose a severe threat to public health in developing nations [1]. Among these, diabetes mellitus stands out as one of the most predominant chronic conditions worlwide, swiftly escalating into a worldwide epidemic [2]. The rising incidence of diabetes mellitus is linked to factors such as population growth, aging, urbanization, increasing rates of obesity, and sedentary behavior [3].

In 1980, According to the World Health Organization (WHO),there were 108 million patients living with diabetes. By 2017, the occurrence of diabetes between adults aged 18–99 years was assessed at 8.4%, with projections indicating a rise to 9.9% by 2045 [4].

The burden of type 2 diabetes mellitus (T2DM) in Iran is substantial, with an overall prevalence of **10.8%** (rising to **21.7% in adults aged 55–64**), disproportionately affecting women (**13.4% vs. 10.8% in men**) and regions like Khuzestan (**15.3%**). Temporal trends show a sharp increase from **7.08% (1988–2002) to 15.0% (2013–2017)**, with obesity (BMI ≥ 35: **19% prevalence**) as a key risk factor. Economically, T2DM costs Iran **152.4 billion PPP (7.69% of GDP)**, with 62% direct costs (medical: 10,819 PPP/patient) and 38% indirect costs, straining healthcare systems (direct medical costs are 6.18× per capita health expenditure). The aging population and complications (e.g., cardiovascular disease) exacerbate disability risks, underscoring the need for targeted prevention, screening, and cost-effective management strategies to mitigate future burdens [5,6].

Besides, Iran faces a significant burden of vitamin D deficiency, with prevalence rates of 59.1% in adults, 76% in adolescents, and 23.3% in infants, far exceeding global levels. Vitamin D deficency contributes to non-communicable diseases, including T2DM, as low vitamin D levels impair insulin sensitivity and increase metabolic dysfunction [7].

The complications of diabetes mellitus tend to progress over time, leading to significant medical expenses, a decline in quality of life, and heightened mortality rates associated with the condition [8]. The vascular and tissue damage resulting from diabetes progression can give rise to severe complications, including retinopathy, nephropathy, cardiovascular disease, cerebral and peripheral vascular disease, and diabetic foot ulcers [9,10].This study primarily concentrates on three major microvascular complications of T2DM and their association with serum vitamin D3 levels: 1) diabetic retinopathy, 2) diabetic neuropathy, and 3) diabetic nephropathy.

1- Diabetic retinopathy is a distinct vascular complication observed in both types of diabetes includingT2DM. Its occurrence is closely linked to the duration of the disease and the effectiveness of glycemic management. As the prominent reason of new cases of adult blindness, diabetic retinopathy poses a significant public health concern [11]. Current projections suggest that the number of individuals affected by this condition will rise to 191 million by 2030 [12].

2- Diabetic neuropathy is among the most prevalent microvascular complications of diabetes, often leading to considerable disability [13]. It can cause severe pain, sensory loss, heightened susceptibility to leg ulcers, diabetic foot, and, in severe cases, amputation [14,15]. The persistent pain associated with this condition significantly impacts patients' sleep, mood, daily functioning, and overall quality of life [16].

3- Diabetic nephropathy is the primary cause of end-stage renal disease globally. Its development is primarily driven by chronic hyperglycemia and hypertension [17]. Early identification and management of these risk factors, along with prompt diagnosis and treatment, are crucial for effective management of the condition [18].

As discussed earlier, the three primary microvascular complications of T2DM are closely linked to blood glucose levels and glycemic control. Additionally, serum vitamin D levels have been presented to influence blood glucose regulation and glycemic control in T2DM. This underscores the importance of understanding the role of vitamin D in these mechanisms. Vitamin D obtained from the skin and diet undergoes metabolism in the liver to form 25-hydroxyvitamin D, which serves as the key indicator of a patient's vitamin D status. Subsequently, 25-hydroxyvitamin D is further metabolized in the kidneys to its active form, 1,25-dihydroxyvitamin D [19]. Receptors for 1,25-dihydroxyvitamin D3 are found in the intestine and bone, as well asin numerous other tissues, such as the brain, heart, stomach, pancreas, activated T and B lymphocytes, skin, and gonads [20]. Animal studies have demonstrated that 1,25-dihydroxyvitamin D3 stimulates pancreatic β-cells to secrete insulin [21]. Findings from numerous animal and human studies indicate that vitamin D may perhaps help reduce the risk of developing diabetes [22].

Vitamin D deficiency is an important health problem that has not yet recognized well [20] and is defined by most experts as a 25-hydroxyvitamin D level of less than 20 ng/ml [19]. Risk factors for vitamin D deficiency include skin pigmentation, use of sunscreen or covering clothing, elderly or being institutionalized, malabsorption, renal and liver disease, obesity, and anticonvulsant drug use [23]. The role of vitamin D deficiency has been recognized as a risk factor for impaired glucose tolerance. As the prevalence of vitamin D deficiency in patients with type 2 diabetes is high, investigation about its' potential adverse effect on diabetic patients are crucial [24]. It is noteworthy that vitamin D deficiency can be treated by giving one dose of 50,000 IU of oral vitamin D once a week for 8 weeks to the patients [25].

Despite numerous studies on the association between vitamin D deficiency and different metabolic diseases, its role in type-2 diabetes was paradoxical. This study aims to investigate the relationship between serum 25(OH) D concentrations and microvascular complications (diabetic nephropathy (macroalbuminuria, and microalbuminuria), retinopathy, neuropathy), and glycemic control in type 2 diabetic patients.

## Materials and methods

### Study design

This analytical cross-sectional study (started: September 16, 2021, and ended: March 23, 2022) includes 199 type-2 diabetic patients who were referred to the endocrinology clinic of Jahrom city. The definitive diagnosis of type 2 diabetes is confirmed and controlled based on updated ADA guideline [26] under the supervision of an endocrinologist (Fig 1). Participants were required to have a confirmed diagnosis of T2DM for at least 1 year prior to enrollment. This criterion ensured access to longitudinal clinical data, including prior HbA1C measurements, complication screenings, and treatment histories, which were essential for analyzing chronic microvascular outcomes. Also it is mportant to note that patients selected for this study were following typical Iranian dietary patterns, which feature a balanced mix of plant-based foods and animal proteins. We did not specifically focus on or recruit vegetarians, vegans, or individuals with restrictive dietary practices.

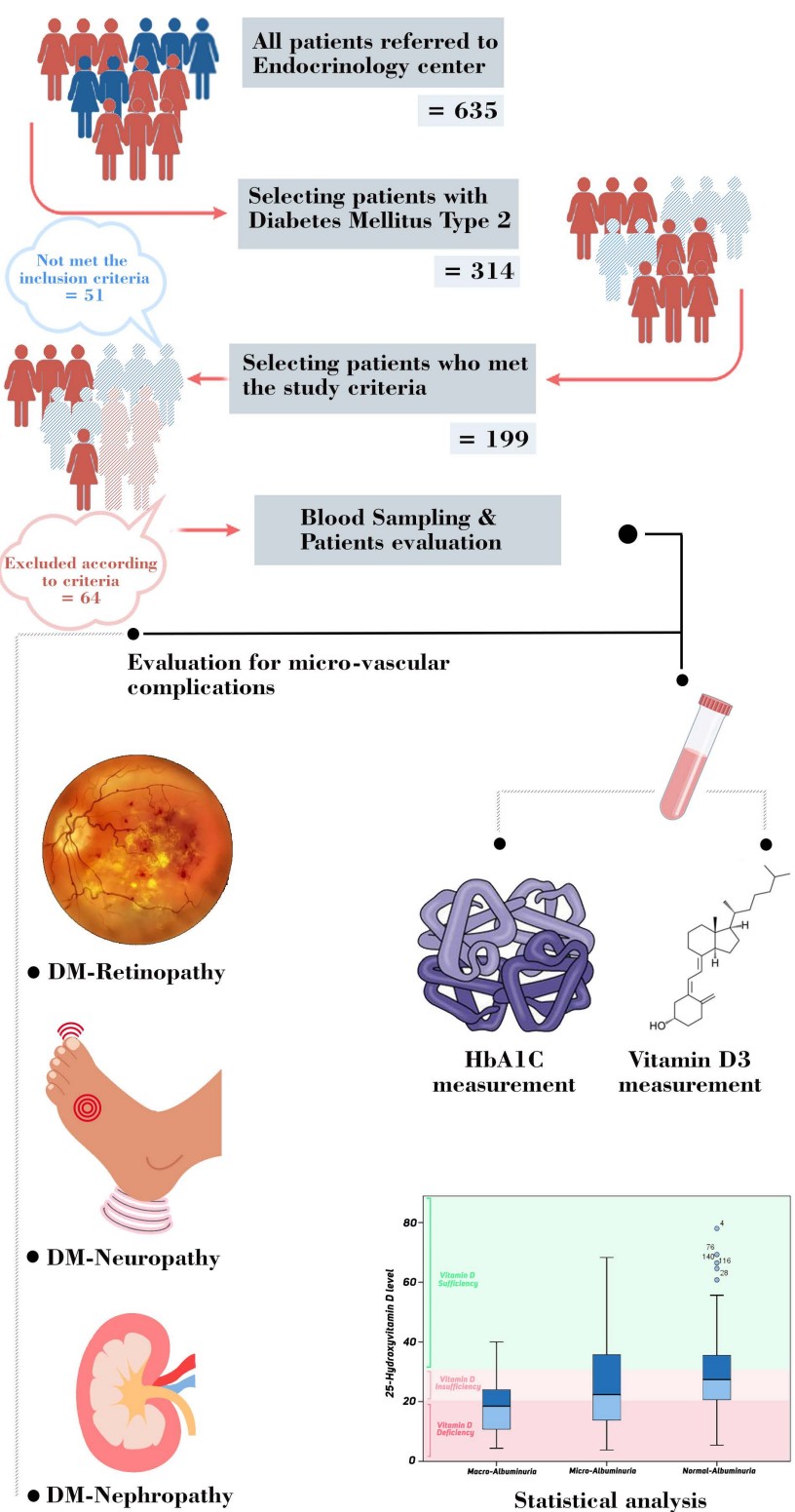

**Fig 1. Graphical abstract of the study design; from patients selection and clinical evaluation and also statistical data analysis.** Exclusion criteria:Type-1 diabetic patients, pregnant or lactating women, patients with malabsorption disorders, celiac disease, inflammatory bowel disease, and those who have undergone gastric bypass surgery or receiving glucocorticoids were excluded.

## Data collection and measurements

Age, sex, weight, height, and period of diabetes were documented for all the subjects. The body mass index (BMI) was calculated as weight (kg) divided by the square of height (m²). Blood pressure (BP) was measured after at least a 10-min rest, with the patient in a seated position (back supported, feet flat, arm at heart level). For patients with elevated readings, two additional measurements were taken at 1–2-minute intervals, and the average was recorded, consistent with WHO/AHA protocols for hypertension diagnosis.

Blood samples were drawn in the morning after at least an 8-h fast and also 2 hours after breakfast. Fasting blood sugar (FBS), 2-hour post-prandial blood glucose (2HPP), glycated hemoglobin $A_1C$ (HbA$_1$C), and serum creatinine were measured. Patients were categorized into three groups based on the HbA$_1$C index (HbA$_1$C < 7.5% or appropriate glycemic control, ≥ 7.5 and < 8% or inappropriate glycemic control, and HbA$_1$C ≥ 8% or uncontrolled [27].

Serum vitamin D$_3$ level was measured by assessing the level of serum 25(OH) vitamin D in samples. This measurement was done by LIAISON vitamin D chemiluminescence immunoassay (DiaSorin, Saluggia, Italy). The serum concentration of ≥ 30 ng/ml was considered sufficient, ≥ 20 and < 30 ng/ml as insufficient, and < 20 ng/ml as deficient [17].

Random urine samples were collected to measure the urine albumin creatinine ratio (UACR). Patients were categorized into three groups based on the UACR. Values ≥ 300 mg/g creatinine were defined as macroalbuminuria, ≥ 300 and < 300 mg/g creatinine as microalbuminuria, and < 30 mg/g creatinine as normoalbuminuria [16]. In patients with UACR ≥ 30 and < 300 mg/g creatinine, random urine samples were repeated at least three more times with intervals of several months. If two out of four samples were ≥ 30 and < 300 mg/g creatinine, the patient was considered to have microalbuminuria [28].

All type 2 diabetic patients were sent to an ophthalmologist as soon as they were diagnosed [29]. Diabetic retinopathy was diagnosed by an ophthalmologist based on funduscopic and slit lamp examination findings and related treatments. Diabetic neuropathy was also diagnosed based on monofilament 10-g, position, vibration, and autonomic tests as long as recording the complaints and history of the patients [30] (Fig 1 & S1 Fig).

## Statistical analysis

The SPSS Statistics 22® (IBM Corp.) program was used for statistical analysis. mean and standard deviation formation was set for variables with normal distribution, while variables with non-normal distribution were arranged in the form of median and 25th – 75th percentile, and nominal variables are expressed as numbers and percentages. The normality statement in variables was calclated using the Kolmogorov–Smirnov test. Distributions higher than P > 0.05 were accepted as normally distributed variables. Kruskal-Wallis tests were used to compare the variables between three subgroups. The chi-square test, Pearson correlation test, and Eta coefficient test were also used for statistical analysis. A p-value less than 0.05 was considered significant.

## Ethical consideration

The study protocol was approved by the research ethics committee of Jahrom University of Medical Sciences (IR.JUMS.REC.1399.154). All patients were voluntarily participate to this study and they signed an inform consent to share their data with us and for publication.

## Results

### Baseline characteristics

The clinical and laboratory characteristics such as sex, age, body mass index (BMI), duration of diabetes, hypertension, systolic and diastolic blood pressure, HbA$_1$C, fasting blood sugar (FBS), 2-hour post-prandial sugar (2HPP), serum creatinine and urine microalbumin to creatinine ratio are as follows (Table 1). Here is to note that baseline characteristics of

**Table 1. Clinical and laboratory characteristics of patients. Cell contents are expressed as a number, percentage, mean ± s.d., or median (25th – 75th percentile). Normally distributed variables are shown as mean ± s.d. nonparametric variables are shown as median (25th – 75th percentile).**

| | Parameter | (N = 199) |
|---|---|---|
| Baseline Characteristics | Sex, F/M | 126/73 |
| | Age, years | 56.79 ± 10.78 |
| | BMI, kg/m² | 28.91 (26.23–32.75) |
| | Duration, years | 8 (3–15) |
| | Hypertension, yes% | 114, 57.3% |
| | SBP, mmHg | 125 (110–140) |
| | DBP, mmHg | 80 (74–82) |
| | HbA$_1$C, % | 7.7 (6.5–8.9) |
| | FBS, mg/dL | 134 (111–173) |
| | 2HPP, mg/dL | 198 (162–263) |
| | SCr, mg/dL | 1 (0.9–1.2) |
| | 25-OHD, ng/ml | 24.3 (18–35.5) |
| | UACR, mg/g | 22 (9.73–78) |
| Microvascular Complications | **Retinopathy, N(percent)** | 51 (25.6%) |
| | **Neuropathy, N(percent)** | 28 (14.1%) |
| | **Microalbuminuria (UACR* 30–300 mg/g)** | 64 (32.2%) |
| | **Macroalbuminuria (UACR ≥ 300 mg/g)** | 16 (8%) |
| Vitamin D deficiency | **Vitamin D Deficiency (< 20 ng/ml)** | 58 (29.1%) |
| | **Vitamin D Insufficiency (20–30 ng/ml)** | 68 (34.2%) |
| | **Vitamin D Sufficiency (≥ 30 ng/ml)** | 73 (36.7%) |

*BMI*, body mass index; *SBP*, systolic blood pressure; *DBP*, diastolic blood pressure; *HbA$_1$C*, hemoglobin A$_1$C; *FBS*, fasting blood sugar; *2HPP*, 2-hour post-prandial blood glucose; *SCr,* serum creatinine; *25-OHD*, 25-hydroxyvitamin D; *UACR*, urine albumin to creatinine ratio. & Frequency of DM microvascular complications. & Frequency of Vitamin D Deficiency, Insufficiency, and Sufficiency.

this study (N = 199) include 56.79 years mean age, 57.3% with hypertension, and 29.1% with vitamin D deficiency. The mean BMI was 28.91 kg/m². The frequency of microvascular complications of type 2 diabetes in a total of 199 established type 2 diabetic patients is shown in the second part of Table 1 (nephropathy (40%), retinopathy (25.6%) and neuropathy (14.1%)). Also, the status of vitamin D levels can be seen in the third part of Table 1, which demonstrates high frequency of vitamin deficiency beside its insufficiency among T2DM patients (63%).

T2DM patients with diabetic retinopathy had meaningfully lower serum 25-OH D concentration (24.59 ± 13.77 ng/ml) and besides, a higher prevalence of vitamin D deficiency and insufficiency (37.25%, 35.3%) in comparison with those without retinopathy(27.96 ± 13.54 ng/ml; 26.35%, 33.78%) Table 2 & Fig 2A.

In addition, the participants with diabetic neuropathy had significantly lower serum 25-hydroxyvitamin D concentration (23.21 ± 15.7 ng/ml) and a much higher prevalence of vitamin D deficiency (50%) as well as lower prevalence of vitamin D insufficiency (32.14%) in comparison with those without neuropathy (27.74 ± 13.22 ng/ml; 25.73%, 34.5%) Table 2 & Fig 2B.

Table 2. Distribution of patients based on the presence or absence of microvascular complications and vitamin D level.

| Vitamin D status | UACR | | | Retinopathy | | Neuropathy | |
|---|---|---|---|---|---|---|---|
| | Macroalbuminuria (UACR* ≥ 300 mg/g) | Microalbuminuria (UACR 30–300 mg/g) | Neg. | No | Yes | No | Yes |
| Vitamin D Deficiency (< 20 ng/ml) | 9 | 24 | 25 | 39 | 19 | 44 | 14 |
| Vitamin D Insufficiency (20–30 ng/ml) | 5 | 19 | 44 | 50 | 18 | 59 | 9 |
| Vitamin D Sufficiency (≥ 30 ng/ml) | 2 | 21 | 50 | 59 | 14 | 68 | 5 |
| Vitamin D level | 19.21 ± 9.76 ng/ml | 24.66 ± 13.54 ng/ml | 29.47 ± 13.6 ng/ml | 27.96 ± 13.54 ng/ml | 24.59 ± 13.77 ng/ml | 27.74 ± 13.22 ng/ml | 23.21 ± 15.7 ng/ml |

*UACR, urine albumin to creatinine ratio.

Furthermore, the patients with macroalbuminuria had significantly lower serum 25-hydroxyvitamin D concentration (19.21 ± 9.76 ng/ml) and a much higher prevalence of vitamin D deficiency (56.25%) than microalbuminuric (24.66 ± 13.54 ng/ml, 37.5%) and normoalbuminuric ones(29.47 ± 13.6 ng/ml, 41.18%) Table 2 & Fig 2C.

As demonstrated in Fig 3A-heatmap there is a significant negative correlation between vitamin D levels and urine albumin creatinine ratio and HBA1C. The outcomes of the Pearson correlation test demonstrated that there is a slight inverse correlation between the two variables of urine albumin creatinine ratio and vitamin D level (25-OHD) (r = -0.175, p = 0.018) Fig 3B. The results of the eta (η) coefficient test showed that the change in vitamin D level (25-OHD) is related to the occurrence of retinopathy, but p.value was not significant (η = 0.903, p = 0.68). Also, the results of the eta coefficient test showed that the change in vitamin D level (25-OHD) is strongly related to the occurrence of neuropathy (η = 0.975, p < 0.001). The results of the Pearson correlation test showed that there is a slight inverse correlation between the two variables of the HbA1C index and vitamin D level (25-OHD) (r = -0.19, p = 0.007) Fig 3C.

The patients with $HbA_1C \geq 8\%$ had significantly lower serum 25-hydroxyvitamin D concentration (24.47 ± 13.82 ng/ml) and higher prevalence of vitamin D deficiency (38.2%) than patients with $HbA_1C$ 7.5% - 8% (26.53 ± 11.19 ng/ml, 26.32%) and patients with $HbA_1C < 7.5\%$ (29.78 ± 13.54 ng/ml, 20.88%) Fig 2D also see Table 3 & S2 and S3 Figs.

## Discussion

The relationship between 25(OH) vitamin D and microvascular complications, align with the glycemic control status of diabetes mellitus type 2 were investigated. The maximal prevalence of microvascular complications: microalbuminuria, retinopathy, neuropathy, and macroalbuminuria was found in the patients with vitamin D deficiency. Poorly controlled diabetes ($HbA_1C \geq 8\%$) was clearly related to lower levels of vitamin D, and patients who had appropriate glycemic control ($HbA_1C < 7.5\%$) had higher serum 25 (OH) D concentrations.

As mentioned previously, patients with retinopathy had a lower mean serum 25(OH) D in comparison with participants without retinopathy. These results were consistent with following previous studies. Afarid et al found that the mean serum 25(OH) D concentration in patients with diabetic retinopathy was lower than in those without diabetic retinopathy [31]. In Luo et al meta-analysis, type 2 diabetes patients with vitamin D deficiency (serum 25(OH) D levels <20 ng/mL) have a significantly increased risk of diabetic retinopathy and an obvious decrease of 1.7 ng/mL in serum vitamin D was established in the patients with diabetic retinopathy [32]. There are also more similar studies aligning with our results [33–36]. Conversely in Alam et al study, there was no difference in serum 25(OH) D between those with and those without diabetic maculopathy [37]. In Bonakdaran et al study, correlation among 25(OH) D level and other recognized risk factors of

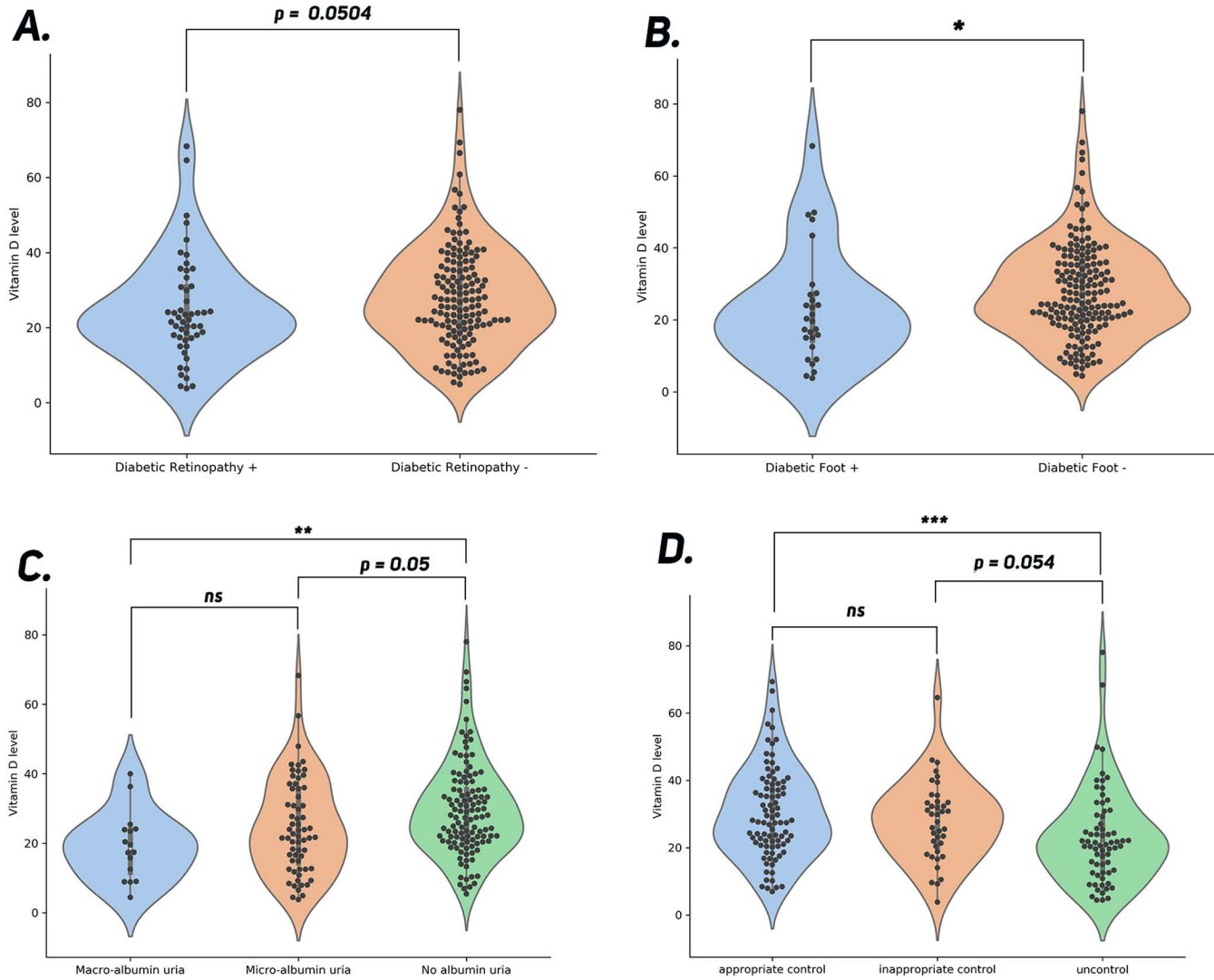

**Fig 2. Comparisons of serum vitamin D levels in different groups of diabetic patients. (A) Vitamin D levels in patients with and without diabetic retinopathy**: The violin plot shows a marginal difference in vitamin D levels between patients with and without retinopathy (p = 0.0504). **(B) Vitamin D levels in patients with and without diabetic foot**: A significant difference is observed, with lower vitamin D levels in patients with diabetic foot compared to those without (p < 0.05). **(C) Vitamin D levels across albuminuria groups**: No significant difference (ns) is seen between macroalbuminuria and microalbuminuria groups, but a significant difference is observed between the no albuminuria group and the other groups (p = 0.05). **(D) Vitamin D levels in control groups based on diabetic control status**: A significant difference (***p < 0.001) is found between uncontrolled and appropriate control groups, with a borderline significant difference between inappropriate control and uncontrolled groups (p = 0.054).

diabetic retinopathy was not significant [38]. Our analysis revealed that there is a significant relationship between serum 25-hydroxyvitamin D level and retinopathy.

Our results showed that a significantly higher prevalence of vitamin D deficiency (50%) was observed in neuropathic diabetic patients in comparison with diabetic participants without neuropathy (25.73%). In He et al cross-sectional study, T2DM patients with diabetic peripheral neuropathy had significantly lesser serum 25(OH) D concentration and also higher prevalence of vitamin D deficiency (80%) than non-diabetic neuropathy patients [39]. According to Niu et al study [40], a serum 25(OH)D level < 34.87 nmol/L proposes the incidence of neuropathy in elderly patients with type 2 diabetes. In

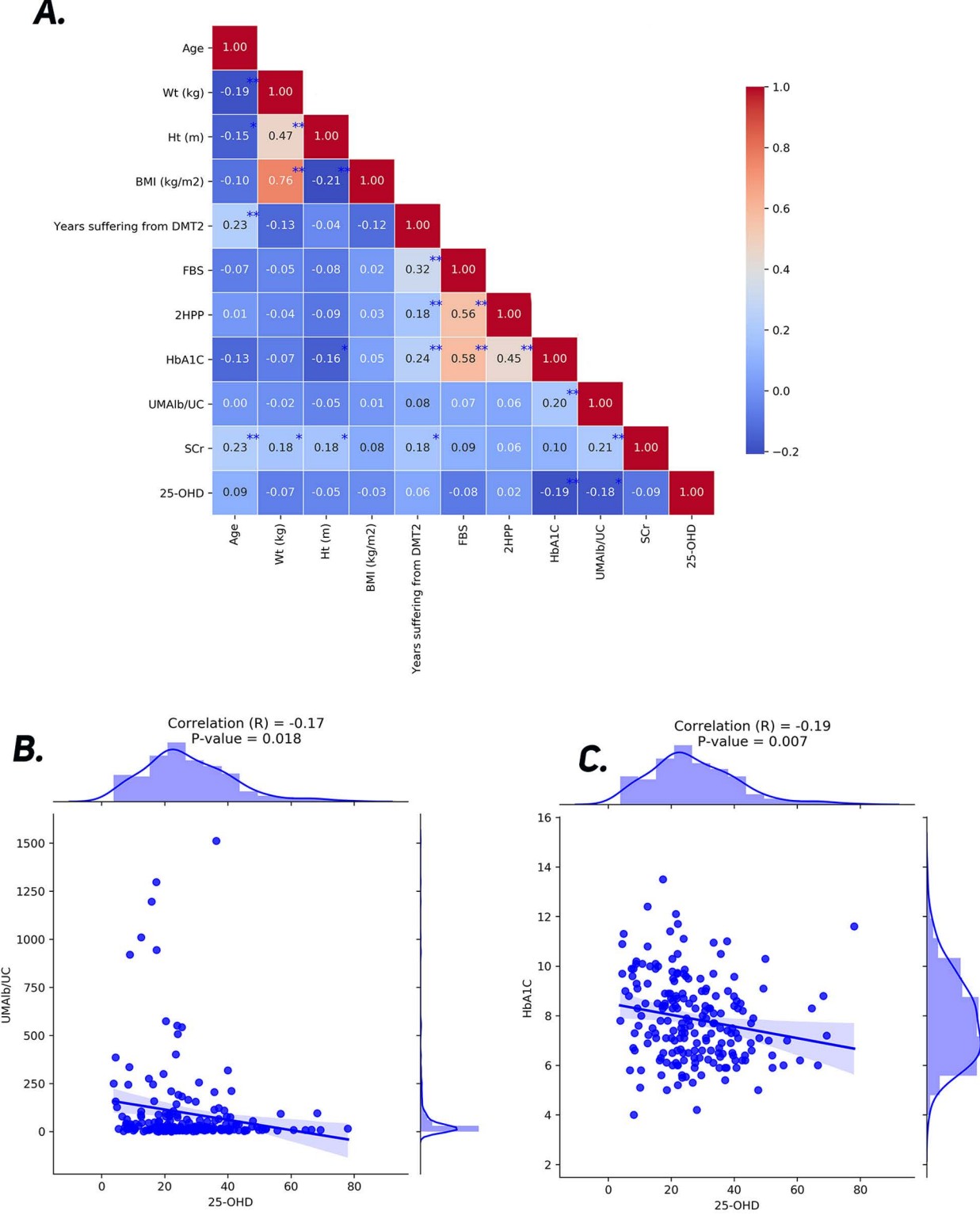

**Fig 3. Correlation of clinical and biochemical parameters; (A) Correlation matrix of clinical and biochemical parameters**: Pearson correlation coefficients (R) between age, weight, height, BMI, duration of type 2 diabetes (DMT2), fasting blood sugar (FBS), 2-hour postprandial blood sugar (2HPP), HbA1C, urinary albumin/creatinine ratio (UMAlb/UC), serum creatinine (S.Cr), and serum 25-hydroxyvitamin D (25-OHD). Significant

correlations are highlighted with asterisks. Strong positive correlations are shown in red, and negative correlations in blue, with color intensity reflecting the strength of the correlation. **(B) Scatter plot with regression line of 25-OHD versus UMAlb/UC**: A weak negative correlation is observed between 25-OHD levels and urinary albumin/creatinine ratio (R = -0.17, p = 0.018), indicating that lower vitamin D levels are associated with higher albuminuria. **(C) Scatter plot with regression line of 25-OHD versus HbA1C**: A weak negative correlation (R = -0.19, p = 0.007) is seen between 25-OHD levels and HbA1C, suggesting that lower vitamin D levels may be linked to poorer glycemic control.

**Table 3. Distribution of patients based on glycemic control and vitamin D level.**

| Vitamin D status | HbA$_1$C* Index | | |
|---|---|---|---|
| | **Appropriate Controlled (HbA$_1$C < 7.5%)** | **Inappropriate Controlled (HbA$_1$C 7.5–8%)** | **Uncontrolled (HbA$_1$C ≥ 8%)** |
| Vitamin D Deficiency (< 20 ng/ml) | 19 | 5 | 34 |
| Vitamin D Insufficiency (20–30 ng/ml) | 33 | 7 | 28 |
| Vitamin D Sufficiency (> 30 ng/ml) | 39 | 7 | 27 |
| Vitamin D level | (29.78 ± 13.54 ng/ml) | (26.53 ± 11.19 ng/ml) | (24.47 ± 13.82 ng/ml) |

*HbA$_1$C, hemoglobin A$_1$C.

Zhang et al meta-analysis [41] the serum concentration of 25(OH) D in type 2 diabetes combined with neuropathy group was lower than in the group without neuropathy. On the contrary, Huang et al results provided no evidence to support the causal association of serum 25 (OH (D levels with diabetic neuropathy (OR = 0.99, 95% CI = 0.98–1.00, P = 0.09) [42]. Our analysis revealed that there is a significant relationship between serum 25-hydroxyvitamin D level and neuropathy.

The serum level of 25-hydroxyvitamin D in patients with macroalbuminuria (19.21 ± 9.76 ng/ml) was lower than patients with microalbuminuria (24.66 ± 13.54 ng/ml) and normoalbuminuric patients (29.47 ± 13.6 ng/ml). In Felicio et al cross-sectional study that included 1576 diabetic patients, The 25(OH) D concentration in patients with normoalbuminuria were higher than the levels detected in those with micro or macroalbuminuria [43]. Also, there was a higher prevalence of vitamin D deficiency (56.25%) in patients with macroalbuminuria than normoalbuminuric patients. However prevalence of vitamin D deficiency in patients with microalbuminuria was the lowest. Özgür et al found that as vitamin D levels decreased, the frequency of albuminuria was on an increasing trend [44]. Our analysis revealed that there is a slight inverse correlation between urine albumin creatinine ratio and serum 25(OH) D concentration. Similar results were found in the other studies. For example, in a meta-analysis by Derakhshanian et al, a significant reverse connotation between serum vitamin D status and also the risk for nephropathy in patients with diabetes was observed [45] and a 25OHD level ≤ 21 ng/ml was considered an optimal cut-off point value for having macroalbuminuria in diabetic patients [46].

Our data demonstrated that higher serum 25 (OH) D concentrations were observed in well glycemic control participants. Our analysis showed that there is a slight inverse correlation between HbA$_1$C index and 25 (OH) D level. Same results was observed in multiple studies [43,47,48]; although some studies found no significant relationship between 25 (OH) D levels and HbA1c [49–51].

In a recent cross-sectional study, Chen et al. (2022) [52] evaluated the link between vitamin D deficiency and microvascular complications in a cohort of T2DM patients from a Chinese population. Their findings demonstrated that lower serum 25(OH)D levels were significantly correlated with a higher prevalence of diabetic retinopathy and nephropathy, even after

adjusting for glycemic control and other metabolic parameters. This observation aligns with our results, where vitamin D deficiency was markedly associated with an increased risk of microvascular complications, particularly in patients with poor glycemic control. The study by Cheng et al. also highlighted that subtle regional and dietary differences might modulate vitamin D status, a point that resonates with our emphasis on the typical Iranian dietary habits of our study [52].

Further reinforcing the therapeutic potential of vitamin D, a systematic review and meta-analysis by Xuan et al. (2022) [53] examined the effect of vitamin D supplementation in patients with diabetic nephropathy. Their analysis of 10 randomized controlled trials, encompassing 651 patients, revealed that vitamin D supplementation significantly increased serum vitamin D levels while concurrently reducing urinary protein excretion and blood creatinine levels. These findings suggest that vitamin D not only acts as a protective factor in diabetic nephropathy but may also ameliorate kidney dysfunction when used alongside standard treatments. Such results complement our own findings, underscoring the clinical relevance of maintaining adequate vitamin D status in mitigating microvascular complications associated with type 2 diabetes.

The main strength of this study is the investigation of the three major diabetic microvascular complications and the simultaneous evaluation of glycemic control correlation with vitamin D in the participants. The small sample size and the cross-sectional design are the limitations of our study. Furthermore, sunlight exposure [54], outdoor activity time, and patients' diet were not considered in the study. The sample size was determined by feasibility, and no formal a priori power calculation was conducted. Post-hoc analyses indicated sufficient power (≥80%) for detecting moderate-to-large effects (e.g., neuropathy-HbA1C correlations) but limited power for smaller associations (e.g., retinopathy). Future studies should prioritize prospective power calculations to validate these findings.

## Conclusion

The consistent relationship of vitamin D levels with diabetic microvascular complications as well as glycemic control opens a new insight into diabetes management for consideration due to the availability and further nutritional importance of vitamin D.

## Supporting information

**S1 Fig. The study design and results of study patients.**
(JPG)

**S2A Fig. Distribution of key baseline characteristics of the study participants (N = 199).** Age Distribution: Displays the frequency of participants by age (mean age = 56.79 ± 10.8 years). BMI Distribution: Shows the frequency of participants by body mass index (mean BMI = 28.91 kg/m²). FBS (Fasting Blood Sugar) Distribution: Depicts the frequency of participants based on their fasting blood sugar levels. HbA1C Distribution: Illustrates the frequency of participants by HbA1C percentage, an indicator of glycemic control.
(JPG)

**S2B Fig. Vitamin D status and its association with health conditions among the participants.** This bar chart shows the counts of patients categorized by vitamin D status (deficiency, insufficiency, and sufficiency) and the presence or absence of specific health conditions, including macroalbuminuria, retinopathy, and neuropathy. Deficiency, insufficiency, and sufficiency are compared for each health condition to highlight the relationship between vitamin D status and the prevalence of microvascular complications.
(JPG)

**S3A Fig. Comparison of Vitamin D Levels According to Glycemic Control: Vitamin D levels are compared across three glycemic control categories: appropriately controlled (HbA1C < 7.5%), inappropriately controlled (HbA1C ≥ 7.5% and < 8%), and uncontrolled (HbA1C ≥ 8%).** The Kruskal-Wallis test was used for statistical analysis.

*p < 0.05 for comparisons between appropriately controlled vs. inappropriately controlled and vs. uncontrolled groups. Though significant, p-values are not shown in the figure.
(JPG)

**S3B Fig. Comparison of Vitamin D Levels According to Albuminuric Stages: Vitamin D levels are compared across three stages of albuminuria: normoalbuminuria (< 30 mg/g creatinine), microalbuminuria (≥ 30 mg/g and < 300 mg/g creatinine), and macroalbuminuria (≥ 300 mg/g creatinine).** The Kruskal-Wallis test was used for statistical analysis. *p < 0.05 for comparisons between normoalbuminuria vs. microalbuminuria and macroalbuminuria. Although significant, p-values are not displayed in the figure.
(JPG)

## Author contributions

**Conceptualization:** Mohammad Aref Bagherzadeh, Zhila Rahmanian.

**Data curation:** Amirreza Reiskarimian.

**Formal analysis:** Mohammad Aref Bagherzadeh.

**Methodology:** Salma Ahi, Mohammad Aref Bagherzadeh, Zhila Rahmanian, Saeed Sobhanian.

**Project administration:** Salma Ahi.

**Supervision:** Salma Ahi.

**Validation:** Mohammad Aref Bagherzadeh.

**Writing – original draft:** Amirreza Reiskarimian, Parisa Pilban.

**Writing – review & editing:** Salma Ahi, Mohammad Aref Bagherzadeh.

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
