## [Decision Letter · Decision Letter 0]

26 Feb 2025

PONE-D-24-48334Impact of Vitamin D on Glycemic Control and Microvascular Complications in Type 2 Diabetes: A Cross-Sectional StudyPLOS ONE

Dear Dr. Ahi,

Thank you for submitting your manuscript to PLOS ONE. After careful consideration, we feel that it has merit but does not fully meet PLOS ONE’s publication criteria as it currently stands. Therefore, we invite you to submit a revised version of the manuscript that addresses the points raised during the review process.

We look forward to receiving your revised manuscript.

Kind regards,

Santhi Silambanan, MD, DNB

Academic Editor

PLOS ONE

Journal Requirements:

- Microvascular Complications and Foot Care: Standards of Medical Care in Diabetes—2021 - https://doi.org/10.2337/dc21-S011

- Ultraviolet Radiation: A Hazard to Children and Adolescents  - https://doi.org/10.1542/peds.2010-3502

In your revision ensure you cite all your sources (including your own works), and quote or rephrase any duplicated text outside the methods section. Further consideration is dependent on these concerns being addressed.

4. In this instance it seems there may be acceptable restrictions in place that prevent the public sharing of your minimal data. However, in line with our goal of ensuring long-term data availability to all interested researchers, PLOS’ Data Policy states that authors cannot be the sole named individuals responsible for ensuring data access (http://journals.plos.org/plosone/s/data-availability#loc-acceptable-data-sharing-methods).

6. We note you have included a table to which you do not refer in the text of your manuscript. Please ensure that you refer to Table 3 in your text; if accepted, production will need this reference to link the reader to the Table.

Reviewers' comments:

Reviewer's Responses to Questions

**Comments to the Author**

1. Is the manuscript technically sound, and do the data support the conclusions?

Reviewer #1: Yes

Reviewer #2: Yes

2. Has the statistical analysis been performed appropriately and rigorously? 

Reviewer #1: Yes

Reviewer #2: Yes

3. Have the authors made all data underlying the findings in their manuscript fully available?

Reviewer #1: No

Reviewer #2: Yes

4. Is the manuscript presented in an intelligible fashion and written in standard English?

Reviewer #1: Yes

Reviewer #2: Yes

5. Review Comments to the Author

Reviewer #1: The author could comment on the burden of type 2 Diabetes mellitus in the study area.

The author could mention the status of Vitamin D Deficiency in the study area.

Methods 2nd paragraph can include exclusion criteria as the subheading.

If the BP was high, the author could mention whether a single reading or a series of readings were taken and the average noted down.

Was there any sample size calculation done? Or is any minimum sample size required calculated?

The author could mention if only newly diagnosed T2DM patients or patients with existing T2DM were included in the inclusion criteria.

Reviewer #2: Please mention what was the basis for the sample size calculation.

What were the food habits of the study participants? Veg/non-veg/Vegan??

Discussion needs more recent article citation and literature description.

6. PLOS authors have the option to publish the peer review history of their article (what does this mean? ). If published, this will include your full peer review and any attached files.

**Do you want your identity to be public for this peer review?** For information about this choice, including consent withdrawal, please see our Privacy Policy .

Reviewer #1: No

Reviewer #2: No

---

## [Author Response · Author response to Decision Letter 1]

12 Apr 2025

Response to Editor

Dear Editorial board of "PLOS ONE"

Following your letter regarding the manuscript " Impact of vitamin D on glycemic control and microvascular complications in type 2 diabetes: A cross-sectional study" submitted to the PLOS ONE for publication, we are sending the rebuttal letter explaining the modifications applied on the manuscript. We have implemented required revisions to the manuscript and we believe that the current revised paper complies with the referee’s constructive remarks. Besides addressing the critiques of the reviewers, we have evaluated the manuscript to add the suggested discussions by the reviewers. We found the excellent comments of the editor and reviewer very constructive and insightful, and we appreciate the time that you and the reviewer have devoted to assess our work. Therefore, all the helpful suggestions and comments have been fully considered. We appreciate the respectful editorial board and the reviewers for devoting their time and providing constructive and thoughtful critiques towards our primary manuscript. The revisions, starting with the last submission, are addressed below.

The responses to the comments of the editor and reviewer can be found as follows.

Journal Requirements:

Comment 1. Please ensure that your manuscript meets PLOS ONE's style requirements, including those for file naming. The PLOS ONE style templates can be found at

Author response - 1: We sincerely appreciate your valuable feedback. In response to your comments, we have carefully revised the manuscript to meet PLOS ONE's guidelines. The following changes have been made:

• The title format has been adjusted to comply with PLOS ONE standards.

• Author names and affiliations have been updated to the requested format.

• Section headings have been modified to align with the journal’s preferred style.

• Figure legends and tables have been repositioned within the text, placed after their first reference, as per PLOS ONE guidelines.

• The abstract has been revised to meet the journal’s requirements.

Comment 2. We noticed you have some minor occurrence of overlapping text with the following previous publication(s), which needs to be addressed:

- Microvascular Complications and Foot Care: Standards of Medical Care in Diabetes—2021 - https://doi.org/10.2337/dc21-S011

- Ultraviolet Radiation: A Hazard to Children and Adolescents - https://doi.org/10.1542/peds.2010-3502

In your revision ensure you cite all your sources (including your own works), and quote or rephrase any duplicated text outside the methods section. Further consideration is dependent on these concerns being addressed.

Author response - 2: We sincerely appreciate your careful attention to this matter. We have thoroughly reviewed the manuscript and addressed the instances of overlapping text with the cited publications. The following corrective actions have been taken:

1. Proper Citation: All relevant sources, including the two referenced publications, have been appropriately cited where necessary.

2. Rephrasing: Any duplicated text outside the Methods section has been either rephrased for originality or properly quoted with attribution.

3. Originality Check: We have ensured that all remaining content is presented in our own words, with proper citations where prior work is referenced.

Comment 3. We note that you have indicated that there are restrictions to data sharing for this study. For studies involving human research participant data or other sensitive data, we encourage authors to share de-identified or anonymized data. However, when data cannot be publicly shared for ethical reasons, we allow authors to make their data sets available upon request. For information on unacceptable data access restrictions, please see http://journals.plos.org/plosone/s/data-availability#loc-unacceptable-data-access-restrictions.

Author response - 3: We appreciate the journal’s policy on data sharing and fully support transparency in research. However, due to the sensitive nature of the data in this study, we are unable to publicly share the dataset for the following reasons:

1. Patient Privacy Concerns: Even after de-identification, the data contain potentially identifiable patient information, making unrestricted sharing ethically problematic.

2. Ethical Restrictions: The Ethics Committee of Jahrom University of Medical Sciences has imposed restrictions on public data sharing to protect participant confidentiality.

Data Access Process:

While the data cannot be made publicly available, qualified researchers may request access through a formal application to: Jahrom University of Medical Science (info@jums.ac.ir)

We have updated the Data Availability Statement in the manuscript to reflect these restrictions and provide clear instructions for data requests.

Thank you for your understanding. Please let us know if further clarification is needed.

Comment 4. In this instance it seems there may be acceptable restrictions in place that prevent the public sharing of your minimal data. However, in line with our goal of ensuring long-term data availability to all interested researchers, PLOS’ Data Policy states that authors cannot be the sole named individuals responsible for ensuring data access (http://journals.plos.org/plosone/s/data-availability#loc-acceptable-data-sharing-methods).

Author Response - 4: Thank you for your guidance. We have updated our data availability statement and provide the following non-author institutional contact and details for long-term data access:

Revised Data Availability Statement:

"Due to ethical restrictions imposed by the Ethics Committee of Jahrom University of Medical Sciences to protect participant confidentiality, the data underlying this study cannot be made publicly available. Qualified researchers may request access to the de-identified minimal dataset by contacting the university's independent Ethics Committee at info@jums.ac.ir (with email title: Access to research data) or +98 (715) 4474992. Requests will be reviewed for compliance with ethical standards and institutional regulations. The data will be stored securely in the university’s institutional repository, which guarantees long-term preservation and accessibility for approved researchers, irrespective of author availability."

Comment 5. PLOS requires an ORCID iD for the corresponding author in Editorial Manager on papers submitted after December 6th, 2016. Please ensure that you have an ORCID iD and that it is validated in Editorial Manager. To do this, go to ‘Update my Information’ (in the upper left-hand corner of the main menu), and click on the Fetch/Validate link next to the ORCID field. This will take you to the ORCID site and allow you to create a new iD or authenticate a pre-existing iD in Editorial Manager.

Author Response - 5: Thank you for your comment. We confirm that the corresponding author’s ORCID iD has been validated in Editorial Manager as required. The steps outlined were followed to ensure compliance:

1. The corresponding author’s ORCID iD (0000-0001-9622-2895) has been updated in the Editorial Manager system.

2. The ORCID record is now fully linked and updated in the submission profile.

We have ensured all requirements are met and appreciate your guidance.

Comment 6. We note you have included a table to which you do not refer in the text of your manuscript. Please ensure that you refer to Table 3 in your text; if accepted, production will need this reference to link the reader to the Table.

Author Response - 6: Thank you for highlighting this oversight. We confirm that Table 3 is now referenced in the final paragraph of the Results section. The manuscript has been updated accordingly.

Comment 7. Please include captions for your Supporting Information files at the end of your manuscript, and update any in-text citations to match accordingly. Please see our Supporting Information guidelines for more information: http://journals.plos.org/plosone/s/supporting-information.

Author Response – 7: Thank you for your feedback. We have added captions for the Supporting Information file at the end of the manuscript and updated all in-text citations to reference these materials appropriately.

Comment 8. Please review your reference list to ensure that it is complete and correct. If you have cited papers that have been retracted, please include the rationale for doing so in the manuscript text, or remove these references and replace them with relevant current references. Any changes to the reference list should be mentioned in the rebuttal letter that accompanies your revised manuscript. If you need to cite a retracted article, indicate the article’s retracted status in the References list and also include a citation and full reference for the retraction notice.

Author Response – 8: Thank you for your valuable comment. We have carefully reviewed the reference list to ensure its completeness and accuracy. We confirm that none of the cited papers in our manuscript have been retracted. All references are current and relevant to the study. Also, the reference list has been verified for proper formatting, completeness, and alignment with in-text citations. No changes were required.

Peer reviewers’ comments and Authors′ responses

Reviewers' comments:

Reviewer's Responses to Questions

Comments to the Author

1. Is the manuscript technically sound, and do the data support the conclusions?

Reviewer #1: Yes

Reviewer #2: Yes

2. Has the statistical analysis been performed appropriately and rigorously?

Reviewer #1: Yes

Reviewer #2: Yes

3. Have the authors made all data underlying the findings in their manuscript fully available?

Reviewer #1: No

Reviewer #2: Yes

4. Is the manuscript presented in an intelligible fashion and written in standard English?

Reviewer #1: Yes

Reviewer #2: Yes

Review Comments to the Author

Reviewer #1:

Author: We sincerely appreciate your insightful feedback and constructive comments, which have been invaluable in enhancing the quality of our manuscript. Your suggestions have significantly strengthened the clarity, rigor, and presentation of our work. We have carefully addressed each of your points.

R1-comment-1: The author could comment on the burden of type 2 Diabetes mellitus in the study area.

Author-Response-1: Thank you for your valuable comment. We have added a paragraph addressing this point (Paragraph 3 of the Introduction).

R1-comment-2: The author could mention the status of Vitamin D Deficiency in the study area.

Author-Response-2: This point has been addressed in paragraph 4 of the introduction. Thank you for your comment.

R1-comment-3: Methods 2nd paragraph can include exclusion criteria as the subheading.

Author-Response-3: Many thanks for noting this issue, we added exclusion criteria as a subheading in the 2nd paragraph of methods.

R1-comment-4: If the BP was high, the author could mention whether a single reading or a series of readings were taken and the average noted down.

Author-Response-4: Thank you for raising this important point. In our study, blood pressure (BP) measurements were performed in accordance with the latest international protocols. For patients with elevated BP, two or more readings were taken at 1–2-minute intervals after the initial 10-minute rest period, and the average of these readings was recorded. This approach aligns with the 2023 World Health Organization(1) (WHO) and American Heart Association(2) (AHA) guidelines, which emphasize the importance of multiple BP measurements to confirm hypertension and reduce variability.

For hypertensive patients or those with high initial readings, according to current guidelines, we rechecked the device and cuff size and also patients’ proper positioning. In addition we asked the patient for out-of-office checking the BP if feasible or another session BP measurement in the office.

We have clarified this methodology in the revised m

---

## [Editor Report · Decision Letter 1]

30 Apr 2025

Impact of Vitamin D on Glycemic Control and Microvascular Complications in Type 2 Diabetes: A Cross-Sectional Study

PONE-D-24-48334R1

Dear Dr. Ahi,

We’re pleased to inform you that your manuscript has been judged scientifically suitable for publication and will be formally accepted for publication once it meets all outstanding technical requirements.

Kind regards,

Santhi Silambanan, MD, DNB

Academic Editor

PLOS ONE
---

## [Editor Report · Acceptance letter]

PONE-D-24-48334R1

PLOS ONE

Dear Dr. Ahi,

I'm pleased to inform you that your manuscript has been deemed suitable for publication in PLOS ONE. Congratulations! Your manuscript is now being handed over to our production team.

Kind regards,

on behalf of

Dr. Santhi Silambanan

Academic Editor

PLOS ONE